# Genome-Wide Identification and Expression Analysis of *AMT* and *NRT* Gene Family in Pecan (*Carya illinoinensis*) Seedlings Revealed a Preference for NH_4_^+^-N

**DOI:** 10.3390/ijms232113314

**Published:** 2022-11-01

**Authors:** Mengyun Chen, Kaikai Zhu, Junyi Xie, Junping Liu, Pengpeng Tan, Fangren Peng

**Affiliations:** 1Co-Innovation Center for Sustainable Forestry in Southern China, College of Forestry, Nanjing Forestry University, Nanjing 210037, China; 2College of Forestry, Nanjing Forestry University, Nanjing 210037, China; 3Department of Ecology, Nanjing Forestry University, Nanjing 210037, China

**Keywords:** NH_4_^+^, NO_3_^−^, AMT, NRT, pecan

## Abstract

Nitrogen (N) is a major limiting factor for plant growth and crop production. The use of N fertilizer in forestry production is increasing each year, but the loss is substantial. Mastering the regulatory mechanisms of N uptake and transport is a key way to improve plant nitrogen use efficiency (NUE). However, this has rarely been studied in pecans. In this study, 10 AMT and 69 NRT gene family members were identified and systematically analyzed from the whole pecan genome using a bioinformatics approach, and the expression patterns of *AMT* and *NRT* genes and the uptake characteristics of NH_4_^+^ and NO_3_^−^ in pecan were analyzed by aeroponic cultivation at varying NH_4_^+^/NO_3_^−^ ratios (0/0, 0/100,25/75, 50/50, 75/25,100/0 as CK, T1, T2, T3, T4, and T5). The results showed that gene duplication was the main reason for the amplification of the *AMT* and *NRT* gene families in pecan, both of which experienced purifying selection. Based on qRT-PCR results, *CiAMTs* were primarily expressed in roots, and *CiNRTs* were majorly expressed in leaves, which were consistent with the distribution of pecan NH_4_^+^ and NO_3_^−^ concentrations in the organs. The expression levels of *CiAMTs* and *CiNRTs* were mainly significantly upregulated under N deficiency and T4 treatment. Meanwhile, T4 treatment significantly increased the NH_4_^+^, NO_3_^−^, and NO_2_^−^ concentrations as well as the Vmax and Km values of NH_4_^+^ and NO_3_^−^ in pecans, and Vmax/Km indicated that pecan seedlings preferred to absorb NH_4_^+^. In summary, considering the single N source of T5, we suggested that the NH_4_^+^/NO_3_^−^ ratio of 75:25 was more beneficial to improve the NUE of pecan, thus increasing pecan yield, which provides a theoretical basis for promoting the scale development of pecan and provides a basis for further identification of the functions of *AMT* and *NRT* genes in the N uptake and transport process of pecan.

## 1. Introduction

Ammonium nitrogen (NH_4_^+^) and nitrate nitrogen (NO_3_^−^) are the two main forms of nitrogen (N) that plants can absorb and use. The absorption, transport, and assimilation of NH_4_^+^ and NO_3_^−^ in plants have been studied extensively. Physiologically, NH_4_^+^ and NO_3_^−^ uptake and translocation systems were identified mainly by root absorption kinetics, which were classified into two types: high-affinity transport systems (HATs) and low-affinity transport pathways (LATs) [1]. Vmax (maximum ion uptake rate) and Km (Mie’s constant) are the two main parameters in the kinetic equation for root nutrient uptake, which can quantitatively characterize the plant uptake of nutrient ions. In general, a larger Vmax value indicates that the plant has a great uptake potential for a certain ion, and the number of the ion transport carrier protein on the cell membrane determines the size of Vmax. Km indicates the affinity between the ion absorbed by the root system and the uptake site (transport carrier), and the greater the affinity, the smaller the Km value [2]. Previous studies have found that plants have a preference for the intake and utilization of NH_4_^+^ and NO_3_^−^; when both N forms are present, plants will preferentially intake and utilize one of them [3]. At the molecular level, many members of the ammonium transporter (AMT) and nitrate transporter (NRT) gene families have been cloned and characterized, and the uptake and utilization of NH_4_^+^ and NO_3_^−^ by plants is regulated by multiple genes in the N transporter protein family [4,5].

AMTs are carrier proteins that actively transport NH_4_^+^ across biological cell membranes, and they were divided into three main subfamilies: AMT/MEP/Rh (Ammonium transporter/Methylamine permease/Rhesus protein) [6]. The AMT family can usually be divided into two subfamilies, namely AMT1 and AMT2 [7]. Both AMT1 and AMT2 show high affinity for NH_4_^+^, but members of the AMT1 subfamily play a more important role in high-affinity NH_4_^+^ uptake [8]. AMT2 has a more complex gene structure and protein profile than AMT1 [9]. The AMT1 subfamily of *Arabidopsis thaliana* has five members, while AMT2 has low homology with the five AMT1 subfamily members and belongs to the MEP subfamily [10]. Expression of *AtAMT1;1* in roots is significantly correlated with N uptake, whereas expression of *AtAMT1;2* in roots is insensitive to changes in N concentration [11]. *AtAMT1;4* mediate the intake of NH_4_^+^ at the pollen plasma membrane [12]. NH_4_^+^ is regulated into the cytoplasm through *AtAMT1;1* and *AtAMT2;1*, which are primarily expressed in leaves [13]. In addition, *AtAMT2;1* may play a role in the transport of NH_4_^+^ from the roots to the ground [10]. A study by Couturier et al. on AMT proteins in poplar (*Populus* L.) found that the expression of *PtrAMT1;2* was influenced by intracellular N concentration. Most PtrAMT1s were preferentially expressed in roots, while most PtrAMT2s were majorly expressed in stems, and *PtrAMT3;1* was expressed only in senescing leaves [14].

The absorption systems, corresponding genes, and regulatory mechanisms of NO_3_^−^ by plants are different from those of NH_4_^+^. The uptake of NO_3_^-^ by plants is the process of pumping protons out of the cell through the H^+^-ATPase in the plasma membrane, creating pH and electrical (ΔΨ) gradients across the plasma membrane that allows the NO_3_^-^ transporter to take up NO_3_^−^ into the cell [15]. Four families of transporters are known to contribute to nitrate uptake and transport in plants: the nitrate transporter protein 1 (NRT1/PTR/NPF), nitrate transporter 2 (NRT2), chloride channel (CLC), and slow anion-associated channel homolog (SLC/SLAH) family [16]. The NRT1 and NRT2 are responsible for LATS and HATS, respectively [17]. Members of the NRT1 subfamily are responsible for transporting NO_3_^−^, hormones, glucosinolates and dipeptides [18]. In *Arabidopsis*, *AtNPF6;3/AtNRT1;1/CHL1* was the first NRT1 subfamily member to be identified and cloned, and this protein is an amphiphilic NRT [19]. Except for *AtNRT1;1*, most NRT1 exhibited low affinity [20]. *NPF4;6/NRT1;2* and *NPF2;7/NAXT1* were also shown to be involved in root NO_3_^−^ uptake, with *NPF4.6* acting on NO_3_^−^ influx [21] and *NPF2.7* involved in NO_3_^−^ efflux [22]. Other NRT1s are mainly related to the internal transport of NO_3_^−^ in processes such as xylem and phloem loading and transport to leaves or seeds [23]. NRT2 belongs to the major facilitator superfamily (MFS) [24], NRT3/NAR2 is a high-affinity NRT [25]. *NRT2.1*, *NRT2.2*, *NRT2.4*, and *NRT2.5* were known to be associated with root NO_3_^−^ influx, and *AtNRT2.1* expression was induced at low NO_3_^−^ levels and repressed at high NO_3_^−^ concentrations [26]. *AtNRT2.4* and *AtNRT2.5* are associated with root NO_3_^−^ absorbance during severe N deficiency [27]. The NRT3 subfamily plays a significant role in NO_3_^−^ transport by regulating the activity of NRT2, but they are not transport proteins themselves [28].

Pecan (*Carya illinoinensis* (Wangenh.) K. Koch) belongs to the Juglandaceae family and is one of the world’s famous nut tree species, indigenous to the United States and Mexico [29]. Pecans have a good development prospect in China because of the high content of various nutrients and the important economic value of the kernels [30]. However, as an important economic tree species introduced for many years in China, pecan still has the problem of insufficient yield, and low nitrogen use efficiency (NUE) is an important factor leading to the underproduction of pecan. The key way to improve NUE is to master the N absorption and utilization pattern of plants and the molecular regulation mechanism, but there is no systematic and in-depth research on pecan in this subject. In this study, we determined the absorption characteristics of NH_4_^+^ and NO_3_^−^ in different N forms. We identified 10 *AMT* and 69 *NRT* gene family members in pecan and classified them into different evolutionary subfamilies. Then, we analyzed their gene structure, replication events and expression patterns to explore their functions in varying N forms, and we provide a theoretical basis for improving the NUE of pecan.

## 2. Results

### 2.1. Identification and Sequence Analysis of AMT and NRT Gene Family Members in Pecan

A total of 11 AMT candidates were identified that contained AMT or AMT-like repeats, and 95 NRT candidates were identified that contained NRT or NRT-like repeats. After validation of AMT and NRT structural domains by Pfam and NCBI CD-search, 10 were identified as *AMT* gene family members, and 69 were identified as *NRT* gene family members. The *AMT* and *NRT* genes were renamed according to the *Arabidopsis* gene names as well as the NCBI blastp results for subsequent analysis. Appendix A provided details of AMTs and NRTs.

Sequence analysis of pecan *AMT* and *NRT* gene family members revealed that the number of exons in AMTs ranged from one to five, and in NRTs from two to nine. The CDS length of AMTs ranged from 471 to 1542 bp, and in NRTs from 1053 to 2826 bp. Most AMTs (9/10) and NRTs (56/69) were stable proteins with a low protein instability index (instability index < 40). GRAVY analysis showed that the hydration of AMT and NRT proteins in pecan was greater than 0, indicating that these proteins are hydrophobic. Subcellular localization predictions showed that most AMTs (9/10) localized to the cell membrane and most NRTs (64/69) localized to the vesicles.

### 2.2. Phylogenetic Analysis of AMTs and NRTs in Different Species

To decipher the evolutionary relationships and functional associations of pecan AMTs and NRTs, phylogenetic trees were constructed using pecan AMT and NRT proteins and other plants, respectively (Figure 1 and Figure 2). According to the phylogenetic trees, all AMT proteins were divided into two distinct evolutionary branches: AMT1 and AMT2 with strong support (Bootstrap = 100%). The AMT2 was further divided into three clusters: AMT2a, AMT2b, and AMT2c. AMT1 was the largest branch and included 34 AMTs (four CiAMTs), AMT2a included two CiAMTs, AMT2b included three CiAMTs, and AMT2c had no pecan *AMT* gene family members.

All NRT proteins were divided into three main clades: NRT1/PTR, NRT2, and NRT3 subfamily. The NRT1 formed four subclasses, named NRT1a, NRT1b, NRT1c, and NRT1d, and included 62 CiNRTs and 52 AtNRTs. NRT2 included five CiNRTs and seven AtNRTs, while NRT3 included two CiNRTs and two AtNRTs. The evolutionary tree had 48 sister pairs, the majority of which were paralogous proteins, 38 pairs in total (21 pairs in pecan and 17 pairs in *Arabidopsis*), and 10 pairs of orthologous proteins. Only the NRT3 subfamily had no orthologous proteins, while all other clades contained orthologous and paralogous proteins.

### 2.3. Phylogenetic Tree, Conserved Motif, Conserved Domain, and Gene Structural Analyses of CiAMTs and CiNRTs

To better understand the evolutionary relationships and functional associations of pecan *AMT* and *NRT* genes, we constructed unrooted phylogenetic trees with pecan AMT and NRT proteins, respectively (Figure 3A and Figure 4A). According to MEME analysis, we found 10 motifs in most of the pecan AMT and NRT proteins (Figure 3B and Figure 4B, Appendix A). Motifs 1 to 7 were found in all AMT subfamilies, suggesting that these motifs may be characteristic motifs associated with members of the *AMT* gene family. Only 2–3 motifs in NRT2 were identical to the NRT1 subfamily, while NRT3 had no identical motifs to the NRT1 subfamily. We analyzed the structural domains of pecan AMT and NRT proteins using Pfam search and found that only one conserved Ammonium_trasp domain existed in all pecan AMT proteins, and they were all located at similar positions, while three conserved structural domains existed in pecan NRT proteins (Figure 3C and Figure 4C, Appendix A). The NRT1 subfamily had two conserved structural domains: PTR2 and MSF_1; NRT2 had only the MSF_1 structural domain; NRT3 contained only the NAR2 structural domain.

In addition, we analyzed the exons/introns of the pecan *AMT* and *NRT* genes to study the structural diversity. The results showed that the AMT1s contained no intron, and the AMT2s contained 2–4 introns (Figure 3D); the NRT1s had 2–9 introns, the NRT2s and NRT3s contained 1–2 introns (Figure 4D). In conclusion, the phylogenetic correlation between gene structure and prediction strongly supports a close evolutionary relationship between paired genes within the same subfamily.

### 2.4. Synteny Analysis of CiAMTs and CiNRTs

To determine the replication events of pecan *AMT* and *NRT* genes, we performed a synteny analysis of the pecan genome. The results showed that there were five duplicated gene pairs among the ten members of the pecan *AMT* gene family, two of which originated from tandem duplication and three from segmental duplication (Appendix A). There are 103 duplicated gene pairs among 69 members of the pecan *NRT* gene family, of which 9 originated from tandem duplication and 94 from segmental duplication. This suggested that fragment replication events played an important role in the expansion of the *AMT* and *NRT* gene families in pecan. 

To examine the selection type of duplicate gene pairs in the pecan *AMT* and *NRT* gene families, the Ka/Ks ratios were analyzed for duplication events (Appendix A). Ka/Ks < 1 means the gene is subjected to purifying selection, Ka/Ks > 1 means the gene underwent positive selection, and Ka/Ks = 1 means neutral evolution. The Ka/Ks values of all duplicate gene pairs were less than 1, indicating that the amplification of the pecan *AMT* and *NRT* genes was mainly influenced by purifying selection.

### 2.5. Effect of N Forms on Quantitative qRT-PCR Analysis of AMT and NRT Gene Expression Levels in Pecan

The results of qRT-PCR analysis of *CiAMTs* showed that the relative expression levels of *CiAMTs* were significantly affected by different N forms (Figure 5). The relative expression levels of almost all *CiAMTs* were higher in roots than in leaves, indicating that *CiAMTs* mainly worked in roots. In the roots, *CiAMT1.1* was significantly upregulated under T3 and T4 (*p* < 0.05), *CiAMT1.3a* was significantly upregulated only under T5 (*p* < 0.05), and *CiAMT1.3b*, *CiAMT1.4* and *CiAMT3.1a* were all significantly upregulated only under T4 (*p* < 0.05). *CiAMT2.1* was significantly upregulated under T1, T3, and T4 (*p* < 0.05), *CiAMT2.2* was significantly upregulated under T4 and T5 (*p* < 0.05), and *CiAMT3.1b* was significantly upregulated under T1, T2, and T3 (*p* < 0.05). The relative expression of *CiAMT3.3* was significantly upregulated under all N form treatments, with the most significant in T4 and T5 (*p* < 0.05). In leaves, all of them showed significant downregulated except *CiAMT2.1*, *CiAMT2.2*, *CiAMT3.1b*, and *CiAMT3.3*, which were significantly upregulated under individual treatments (*p* < 0.05).

Based on the results of previous studies [16], some members of the *Arabidopsis* and rice *NRT* gene families have been shown to be associated with NO_3_^−^ uptake and transport. According to the phylogenetic tree, we selected 16 pecan CiPawNRTs corresponding to them and further analyzed them using qRT-PCR (Figure 6). The qRT-PCR results showed that *CiNRTs* showed different expression patterns in different N forms as well as in different pecan organs. Except for *CiNPF2.13*, *CiNPF4.6*, *CiNPF5.5*, *CiNPF6.3a*, *CiNPF6.4*, *CiNRT2.5*, and *CiNRT2.7*, which showed relative higher expression in leaves than in roots under most N form treatments, most of the other *CiNRTs* showed higher expression in roots than in leaves. In leaves, *CiNPF2.4* and *CiNPF2.13* under T1, *CiNPF5.5* under T1 and T4, *CiNPF7.2* under T3, *CiNRT2.7* under T1, and *CiNPF2.7b*, *CiNPF4.6*, *CiNPF5.9a*, *CiNPF5.10b* under T5 showed significantly upregulated expression (*p* < 0.05). In roots, *CiNPF1.2a* was significantly upregulated under T3 and T4 (*p* < 0.05), and *CiNPF2.7b* was significantly upregulated under T2 and T4 (*p* < 0.05). *CiNPF2.4* and *CiNPF2.11b* were significantly upregulated only with T3 (*p* < 0.05), while *CiNPF5.10b* was significantly upregulated only with T4 (*p* < 0.05). *CiNPF2.13*, *CiNPF6.4,* and *CiNRT2.7* were significantly downregulated under each N form treatment (*p* < 0.05), *CiNPF4.6* was significantly downregulated under T1, T2, and T5 (*p* < 0.05), while *CiNPF7.3a* was significantly downregulated under T1, T2, T3, and T5 (*p* < 0.05). *CiNPF5.5* was significantly upregulated under T2 as well as significantly downregulated under T3 (*p* < 0.05), *CiNPF6.2* was significantly upregulated under T2 as well as significantly downregulated under T1, T3, and T5 (*p* < 0.05), and *CiNPF6.3a* was significantly upregulated under T4 as well as significantly downregulated under T1, T2, and T3 (*p* < 0.05). *CiNPF7.2* was significantly upregulated under T3 and T4 and downregulated under T5 (*p* < 0.05). *CiNRT2.5* was significantly upregulated under T2 and T3 and downregulated under the other treatments (*p* < 0.05).

### 2.6. Effect of N Forms on NH_4_^+^, NO_3_^−^ and NO_2_^−^ Concentration in Pecan

To further investigate the response mechanisms of pecan *AMT* and *NRT* genes to N forms, we measured the concentrations of NH_4_^+^, NO_3_^−^, and NO_2_^−^ in pecan under different N forms (Figure 7). The results showed that there was no significant difference in the NH_4_^+^ concentrations of pecan in all organs under different N forms. Except for the T5 treatment, NH_4_^+^ concentrations under all other treatments showed greater in roots than in leaves and stem (*p* < 0.05) and no significant difference between leaves and stems. The variability of NO_3_^−^ concentrations in each organ of pecan varied among treatments, and T5 showed significantly greater than CK and T3 in leaves (*p* < 0.05), and no significant differences were found between T1, T2, T4, and other treatments. In the stems, no significant differences were found between treatments. In the roots, it showed that T4 was significantly greater than CK (*p* < 0.05), and T4 was not significantly different from the other treatments. Except for T3 and T5 treatment, the NO_3_^−^ concentrations of pecan under all other treatments showed greater leaves than roots (*p* < 0.05) and no significant difference between stems, leaves, and roots. There was no significant difference in NO_2_^−^ concentrations in all organs of pecan under different treatments, while the variability of NO_2_^−^ concentrations in different organs under each treatment varied. CK, T1, and T5 showed no significant difference between stems and roots, and both of them were significantly greater than leaves (*p* < 0.05), T2 and T3 showed no significant difference between leaves and roots, and both of them were significantly greater than stems (*p* < 0.05), while T4 showed no significant variation in different organs.

At the mean level, pecan NH_4_^+^ concentrations were significantly greater in the T4 than in the other treatments (*p* < 0.05), with no significant differences between the other treatments. The pecan NO_3_^-^concentrations revealed no significant difference between T1, T4, and T5, and both of them were significantly greater than CK, T2, and T3 (*p* < 0.05). There was no significant difference between T2 and T1, T3 and T4, T2 was significantly greater than CK (*p* < 0.05), and no significant difference between CK and T3. The variability of pecan NO_2_^-^ and NH_4_^+^ concentrations was consistent, suggesting that T4 was significantly greater than the other treatments (*p* < 0.05), with no significant differences between the other treatments.

### 2.7. Effect of N Forms on the Uptake Kinetics of NH_4_^+^and NO_3_^−^ in Pecan

We also determined the kinetic properties of pecan NH_4_^+^ and NO_3_^−^ uptake under different N forms (Figure 8). The results showed that the uptake rates of NH_4_^+^ and NO_3_^−^ under T2, T3, and T4 were still in a significant upward trend at the ion concentration of 2000 μmol·L^−1^; the uptake rates of NH_4_^+^ and NO_3_^−^ under CK, T1, and T5 leveled off at the medium ion concentration of 1000 μmol·L^−1^.

The root uptake rates were processed data according to the Hofstee transformation equation to obtain the maximum uptake rate (Vm) and the Mee’s constant (Km) of pecan for different N forms with significant coefficients of determination R^2^ (Table 1). Different N forms showed significant effects on Vmax and Km of NH_4_^+^ and NO_3_^−^ (*p* < 0.05), both showing T4 > T3 > T2 > CK > T1 > T5. This indicates that the involvement of a certain proportion of NH_4_^+^ accelerated the uptake of NH_4_^+^ and NO_3_^−^ by pecans, but the affinity decreased, while NH_4_^+^ above a certain proportion decreased the uptake rate and increased the affinity. Except for the T1 treatment, the Vmax/Km of NH_4_^+^ was greater than that of NO_3_^−^ under all treatments, indicating that the rate of NH_4_^+^ uptake by pecan was greater than that of NO_3_^−^.

## 3. Discussion

### 3.1. Functional Differentiation of AMT and NRT Gene Family in Pecan Genome

The biological functions of *AMT* and *NRT* gene families in pecan are poorly understood. Therefore, we report for an earlier time identification of 10 *AMT* and 69 *NRT* gene family members from pecan, confirming that direct orthologs of AMT and NRT proteins should be highly conserved evolutionarily throughout the plant. 

Studies on *Arabidopsis* AMT proteins have shown that AtAMTs had a prominent role in NH_4_^+^ assimilation at the cell membranes [10,11,31], and studies of flowering Chinese cabbage (*Brassica campestris*) also indicated that BcAMT2 was located on the plasma membrane [32], which was consistent with the predicted results of subcellular localization in this study (Appendix A), and this is the optimal cellular structure for maintaining stable NH_4_^+^ concentrations in plants. AtNPF5.11, AtNPF5.12, and AtNPF5.16 were identified to function in the process of NO_3_^−^ from the vesicle to the cytoplasm, thereby regulating NO_3_^−^ distribution between roots and shoots [33]. In contrast, the predicted subcellular localization of the pecan *NRT* genes in this study showed that most of the NRT1s were localized to the vacuoles (Appendix A), suggesting that pecan NRT1s may mostly act in the transport of NO_3_^−^ between the vacuoles and the cytoplasm. There were also some NRT1s localized on the cell membrane, while all NRT2s were all localized to the cell membrane, suggesting that this fraction of NRT proteins may mediate assimilation and efflux of NO_3_^−^ in plants as well as inter-subcellular transport, which was consistent with the findings of cassava (*Manihot esculenta*) [34]. According to the predicted results of subcellular localization, the subcellular localization of all NRT3s differed significantly from NRT1 and NRT2 in pecan, with CiNRT3.1 localized to the nucleus and CiNRT3.2 localized on the cell membrane, cell wall, chloroplast, and vacuoles (Appendix A), indicating that the pecan NRT3s had more complex structures and functions.

### 3.2. Effect of N Forms on the Absorption Characteristics of NH_4_^+^and NO_3_^−^ in Pecan

NH_4_^+^ and NO_3_^−^ are the two main forms of N absorbed and utilized by plants, and previous studies have shown that a nutrient mixture of NH_4_^+^ and NO_3_^−^ can improve crop yield and quality compared to a single source of N [35]. Therefore, understanding the best NH_4_^+^:NO_3_^−^ ratios provides the possibility to improve the N utilization efficiency of plants. We investigated the effect of varying NH_4_^+^:NO_3_^−^ ratios on the absorption and transport of N by measuring the concentrations of different N forms in the pecan organs (Figure 7).

This study showed that the concentrations of different N forms in pecan were tissue-specific. The NH_4_^+^ concentrations of pecan were significantly higher in roots than in leaves and stems, while NO_3_^−^ and NO_2_^−^ concentrations were significantly higher in leaves than in stems and roots, which was consistent with the finding that NH_4_^+^ was majorly assimilated in roots and NO_3_^−^ was mostly translocated to leaves for storage or assimilation [36]. T4 significantly increased the total NH_4_^+^ concentrations of pecan, primarily in the roots, suggesting that T4 was more favorable to promote NH_4_^+^ absorbed by pecan roots, but had no effect on NH_4_^+^ transport between organs. All NH_4_^+^:NO_3_^−^ ratio treatments increased the total NO_3_^−^ concentrations, primarily in the leaves, indicating that the feeding of NH_4_^+^ and NO_3_^−^ mainly promoted the translocation of pecan NO_3_^−^ from roots to leaves, and possibly the acclimation of NO_3_^−^ in the leaves. Compared to T1, T3 reduced the total pecan NO_3_^−^ concentrations, which was in agreement with the results of studies in pepper (*Capsicum annuum*) [37]. T4 significantly increased the total NO_2_^−^ concentrations of pecan, indicating that T4 promoted the conversion of NO_3_^−^ to NO_2_^−^.

Previous conclusions on the effects of the simultaneous presence of NH_4_^+^ and NO_3_^−^ on each other’s uptake were varied, with some suggesting that they have a facilitative or inhibitory effect [38,39], and that plant absorption of NH_4_^+^ and NO_3_^−^ is limited by the maximum uptake threshold [40]. In this study, the uptake rates of NH_4_^+^ and NO_3_^−^ gradually saturated with increasing substrate concentrations under a single N source, while they remained on an increasing trend under mixed N source treatments, with the T4 treatment being the most obvious, indicating that NH_4_^+^ and NO_3_^−^ promoted each other’s intake in this study, as evidenced by the magnitude of Vmax values of pecan. However, single N source treatment increased the affinity of NH_4_^+^ and NO_3_^−^, and the combined application of NH_4_^+^ and NO_3_^−^ decreased their affinity instead, which was generally consistent with the results of Kamminga-Van Wijk and Prins [41]. Vmax/Km is also commonly used to indicate plant preference for NH_4_^+^ and NO_3_^−^ uptake [42], suggesting that pecans may be more biased toward NH_4_^+^ absorption.

### 3.3. Effect of N Forms on the Expression of CiAMTs and CiNRTs

The ingestion and transport of NH_4_^+^ and NO_3_^−^ in plants is majorly mediated by *AMT* and *NRT* gene family members. Therefore, we investigated the expression modes of these genes in varying organs and N forms to further understand the effect of N forms on N uptake and transport in pecan.

In the present study, the relative expression levels of almost all *CiAMTs* were higher in roots than in leaves, indicating that the addition of NH_4_^+^ and NO_3_^−^ majorly stimulated the expression of *CiAMTs* in roots. Studies have shown that AtAMT1.1, AtAMT1.2, AtAMT1.3, and AtAMT1.5 are the principal transporter proteins that take up high-affinity NH_4_^+^ into *Arabidopsis* roots, with AtAMT1.1 and AtAMT1.3 responsible for approximately two-thirds of the high-affinity NH_4_^+^ uptake capacity in the root; the CiAMT1s may have the same function [31,43]. The results of this study showed that all *CiAMT1s* were upregulated in roots of T4, except for *CiAMT1.3a*, which was upregulated in roots under T5, suggesting that T4 may have improved the absorption of high-affinity NH_4_^+^ by the pecan root. In this study, *CiAMT1s* were mostly upregulated in leaves under N lack, but numerous studies have shown that expression levels of *AMT1s* were upregulated in roots under N limited conditions [14,43,44]. This may be due to the variation caused by the longer duration of N deficiency in this study, or it is possible that the expression of *CiAMTs* in leaves was not only affected by N lack, but may also be involved in some other regulatory mechanisms [45]. Previous studies have shown that peach (*Prunus persica*) PpeAMT3;4 was primarily expressed in roots [46], and the expression pattern of *CiAMT2s* was similar to that of *CiAMT1s*, being expressed mostly in roots and almost significantly upregulated at all timepoints under T4.

Studies on *Arabidopsis* suggested that AtNPF6.3/AtNRT1.1 was not only an amphiphilic NO_3_^−^ transport protein but might also act as a NO_3_^−^ sensor under low NO_3_^−^ conditions [47]. The expression level of *CiNPF6.3a* was significantly upregulated in roots under T4, suggesting that it may carry both translocation and signaling functions at this time. AtNPF4.6/AtNRT1.2 was not only a low-affinity NO_3_^−^ transporter protein that plays a role in NO_3_^−^ influx [21], but also an abscisic acid (ABA) transporter protein that positively regulated the ABA response [48]. *CiNPF4.6* expression was significantly upregulated in roots under T3 and in leaves under T5, perhaps because CiNPF4.6 worked primarily on NO_3_^−^ influx under T3, while it worked mostly on ABA regulation under T5. MtNPF6.8/MtNRT1.3 of *Medicago truncatula* was an amphipathic NO_3_^−^ transport protein and was upregulated by the absence of NO_3_^−^ [49]. This was the same as the results of our study, where *CiNPF6.4/CiNRT1.3* expression was significantly upregulated in roots under CK and in leaves under T5. AtNPF7.3/AtNRT1.5 mediated root-to-stem transport, and the same conclusion was found for ZxNPF7.3/ZxNRT1.5 in the *Zygophyllum xanthoxylum*, which also contributed to the uptake of NO_3_^−^ [50]. The expression levels of *CiNPF7.3a*/*CiNRT1.3a* were significantly upregulated in roots, leaves under CK, and roots under T4, indicating that both N scarcity and T4 may promote NO_3_^−^ transport from roots to stems. Moreover, AtNPF7.2/AtNRT1.8 was phylogenetically similar to AtNPF7.3 [51], while *CiNPF7.2*/*CiNRT1.8* was also significantly upregulated in roots under T4 and may function similarly to CiNPF7.3. NPF2.13/NRT1.7 and NPF1.2/NRT1.11 were proven to be involved in the transfer and redistribution of NO_3_^−^ from xylem or the NO_3_^−^ containing tissues to the phloem [23], *CiNPF2.13* was largely induced by N limitation and T3, while *CiNPF1.2a* was mainly induced by N limitation, and T4. NPF2.11/NRT1.10 was identified to be involved in thioglucoside transport [52], and *CiNPF2.11* expression was significantly upregulated under CK, suggesting that CiNPF2.11 may resist N deficiency stress by regulating the concentrations of thioglucosides. AtNRT2.5 was a plasma-membrane localized high-affinity NO_3_^−^ transporter protein that mediated NO_3_^−^ acquisition, and reactivation under N deficient conditions [27], and CiNRT2.5 in this study exhibited the same expression pattern. The transcript levels of NRT2.7 in *Fraxinus mandshurica* were both upregulated in leaves due to N limitation [53], whereas *CiNRT2.7* in this study was significantly expressed in roots under CK and in leaves under T1, which may be caused by species differences.

## 4. Materials and Methods

### 4.1. Plant Materials and Experimental Design

In the study, aeroponic cultivation trials were carried out in the greenhouse of the campus of Nanjing Forestry University from 18 April to 9 June 2021 and from 4 May to 21 June 2022. The plant materials and experimental design refer to Chen et al. [29]. In the case of the same N supply, the five ammonia-to-nitrate ratios (NH_4_^+^:NO_3_^−^) were 100:0, 75:25, 50:50, 25:75, and 0:100, corresponding to T1, T2, T3, T4, and T5, respectively. The nutrient solution without N was used as the control (CK), and each treatment was repeated 3 times, each with 6 seedlings. Regulation of the NH_4_^+^:NO_3_^−^ ratios for each treatment was achieved with specific source compounds (Appendix A). Samples were taken after 45 days of treatment for further determination.

### 4.2. Identification of AMT and NRT Genes in Pecan

To identify the pecan *AMT* and *NRT* genes, we obtained all of the protein sequences of pecan from the Phytozome v13 database (https://phytozome-next.jgi.doe.gov/info/CillinoinensisPawnee_v1_1 (accessed on 1 November 2021)). The hidden Markov model (HMM) profiles of the *AMT* domain (PF00909) and *NRT* domain (PF07690, PF00854, PF16974), downloaded from the Pfam database (http://pfam.xfam.org/ (accessed on 1 February 2022)) [54], were used to identify the pecan AMT and NRT proteins by using HMM search through HMMER3.0 program (www.hmmer.org (accessed on 1 February 2022)) with default parameters [55]. As multiple AMT and NRT proteins were corresponding to a specific gene in several cases, only one protein sequence corresponding to each gene was retained for further detailed analysis. The Pfam database was used to confirm the presence of these conserved domains of the screened genes. The biophysical properties such as amino acid length (AA), molecular weights (MWs), theoretical isoelectric points (pIs), and grand average of hydration (GRAVY) of pecan AMT and NRT proteins were estimated by ExPASy ProtParam server (http://web.expasy.org/protparam (accessed on 1 February 2022)) [56]. Transmembrane helices (TMHs) were determined using the TMHMM tools (http://www.cbs.dtu.dk/services/TMHMM-2.0/ (accessed on 1 February 2022)), prediction of subcellular localization information for pecan AMT and NRT proteins using the Cell-PLoc 2.0 software (http://www.csbio.sjtu.edu.cn/bioinf/Cell-PLoc-2/ (accessed on 1 February 2022)) [57].

### 4.3. Phylogenetic Analysis

The *Arabidopsis* AMT and NRT protein sequences were downloaded from the TAIR database (https://www.arabidopsis.org/ (accessed on 1 November 2021)) [58]. The poplar, apple, pear, tomato, and rice AMT protein sequences were downloaded from the Phytozome database. Phylogenetic trees were constructed by MEGA 7.0 with the following settings: the neighbor-joining (NJ) method, 1000 bootstrap replicates, the Jones–Taylor–Thornton (JTT) model, and pairwise deletion [59]. The evolutionary tree was visualized using the online tool Evolview (http://evolgenius.info/ (accessed on 1 March 2022)) [60]. Because *CiAMT3.1c* is a partial sequence, it was not included in the phylogenetic tree.

### 4.4. Analysis of Gene Structure, Conservative Motifs, and Domains

The pecan genome sequence files and the General Feature Format (GFF) annotation files were downloaded from the Phytozome database. We used the online MEME tool (http://MEME-suite.org/ (accessed on 1 April 2022)) for topic prediction, keeping the maximum number of topics at 10 [61]. TBtools visualizes *AMT* and *NRT* gene structures, conservative motifs, and domains [62].

### 4.5. Synteny Analysis

For detecting syntenic blocks, the whole genome sequence file and the GFF annotation file of pecan were used to identify all duplication events in the pecan genome using MCScanX software [63]. Then, the *AMT* and *NRT* gene families were analyzed for synteny and visualized using TBtools.

### 4.6. Estimation of the Ka/Ks Values

Multiple sequence alignment of full-length coding sequences (CDS) in the *AMT* and *NRT* gene families in pecan was performed using MEGA 7 software and further used to calculate nonsynonymous (Ka) and synonymous (Ks), and the Ka/Ks ratios. Ks values were commonly used to determine the time since gene duplication, and the selection pressure of duplication event was determined by the Ka/Ks ratio.

### 4.7. Cis-Regulatory Elements Analysis

To investigate potential cis-regulatory elements in the promoters of the pecan *AMT* and *NRT* genes, a 1000 bp region upstream of the *AMT* and *NRT* genes was retrieved from the pecan genome sequences. Then, the cis-regulatory elements were predicted using PlantCARE software (//bioinformatics.psb.ugent.be/webtools/plantcare/html/ (accessed on 1 April 2022)) [64] and screened and visualized by TBtools software (Appendix A).

### 4.8. Protein–Protein Interaction Network Prediction

Protein–protein interaction networks of *AMT* and *NRT* gene families were analyzed by String (https://string-db.org/ (accessed on 1 April 2022)) (Appendix A) [65].

### 4.9. RNA Collection and qRT-PCR Expression Analysis

According to the manufacturer’s protocol, the total RNA was extracted from the leaves and roots of pecan using a Universal Plant Total RNA Extraction Kit (Bioteke, Beijing, China) and stored at −80 °C until further use. The purity and integrity of the isolated total RNA were analyzed by agarose gel electrophoresis and Nanodrop 2000 spectrophotometer (Thermo Scientific, Wilmington, NC, USA). First-strand cDNA was synthesized using a cDNA Synthesis Kit (HiScript ^®^RIII RT SuperMix for qPCR +gDNA wiper, Vazyme, Nanjing, China). The qRT-PCR was performed on a 7500 Real-Time PCR system (Applied Biosystems^TM^, Foster City, CA, USA) using a Taq Pro Universal SYBR qPCR Master Mix (Vazyme, Nanjing, China). The specific primers were synthesized by Tsingke Biotechnology Ltd. (Nanjing, China), and the details of the primers were provided in Appendix A. The PCR parameters applied here were as follows: 95 °C for 30 s, followed by 40 cycles of 5 s at 95 and 30 s at 60 °C. The *Actin* gene was used as an internal reference gene [66], and the relative expression levels of pecan *AMT* and *NRT* genes were determined using the 2^−ΔΔCt^ method [67]. Values represent mean calculated from three biological replicates and three technological repeats.

### 4.10. Measurement of NH_4_^+^, NO_3_^−^ and NO_2_^−^ Concentration in Pecan

The NH_4_^+^ concentrations of pecan roots, stems and leaves were determined according to the Berthelot reaction [68]. NO_3_^−^ concentrations were determined according to the method suggested by Patterson et al. [69]. The NO_2_^−^ concentrations were determined by the method of Ogawa et al. [70].

### 4.11. Kinetic Characterization of NH_4_^+^ and NO_3_^−^ Uptake in Pecan

The kinetic characteristics of NH_4_^+^ and NO_3_^−^ uptake in pecan seedlings were determined by the conventional depletion method, the ion concentrations in the solutions to be measured were determined after preparing different concentrations of NH_4_^+^ and NO_3_^−^ for 24 h incubation of the plants, the net rate of ion uptake per unit fresh root per unit time was calculated, and the kinetic parameters of uptake were mathematically derived. The concentration of NH_4_^+^ and NO_3_^−^ was determined by referring to 2.10.

### 4.12. Data Analysis

Before analysis of variance (ANOVA), data were checked for normality and homogeneity of variances. One-way ANOVA was performed to test the effects of different N forms on NH_4_^+^, NO_3_^−^, NO_2_^−^ concentrations, absorption kinetic characteristics and relative gene expression of pecan seedlings. Differences were considered significant at *p* < 0.05.

The kinetic parameters of NH_4_^+^ and NO_3_^−^ uptake were calculated using the Hofstee transformation of the Michaelis–Menten kinetic equation: V = C × Vmax/(Km + C), C represents the ion concentration, V indicates the net ion uptake rate, Vm indicates the maximum uptake rate, and the Km value represents the root uptake site for ion affinity. Non-linear regression fitting and graphing were performed using SPSS to obtain the Vmax and Km. The α value represents the competitive ability of the plant root system for nutrient uptake, α = Vmax/Km.

All statistical analyses were performed with SPSS 23.0 software (Version 23.0, Chicago, IL, USA). All charts were drawn with Excel (Version 2019, Redmond, WA, USA) and SigmaPlot (Version 14.0, Barcelona, Spain).

## 5. Conclusions

We identified 10 *AMT* and 69 *NRT* genes in the pecan genome, and the analysis showed that the biophysical properties, gene structure, and expression levels of *CiAMTs* and *CiNRTs* were strongly associated with pecan NH_4_^+^ and NO_3_^−^ uptake. Combining the effects of different N form treatments on the expression levels of *CiAMTs* and *CiNRT*s, the N concentrations of pecan, and the uptake rate of NH_4_^+^and NO_3_^−^, we concluded that pecan preferred NH_4_^+^ and that the NH_4_^+^/NO_3_^−^ ratio of 75:25 was more favorable to improve the N uptake capacity of pecan seedlings. This study provides a basis for further identification of the functions of *AMT* and *NRT* genes in N uptake and transport of pecan, and it provides a theoretical basis for the application of an optimal proportion of N fertilizer to improve NUE, thereby increasing pecan yield and promoting pecan industrialization.

## Figures and Tables

**Figure 1 ijms-23-13314-f001:**
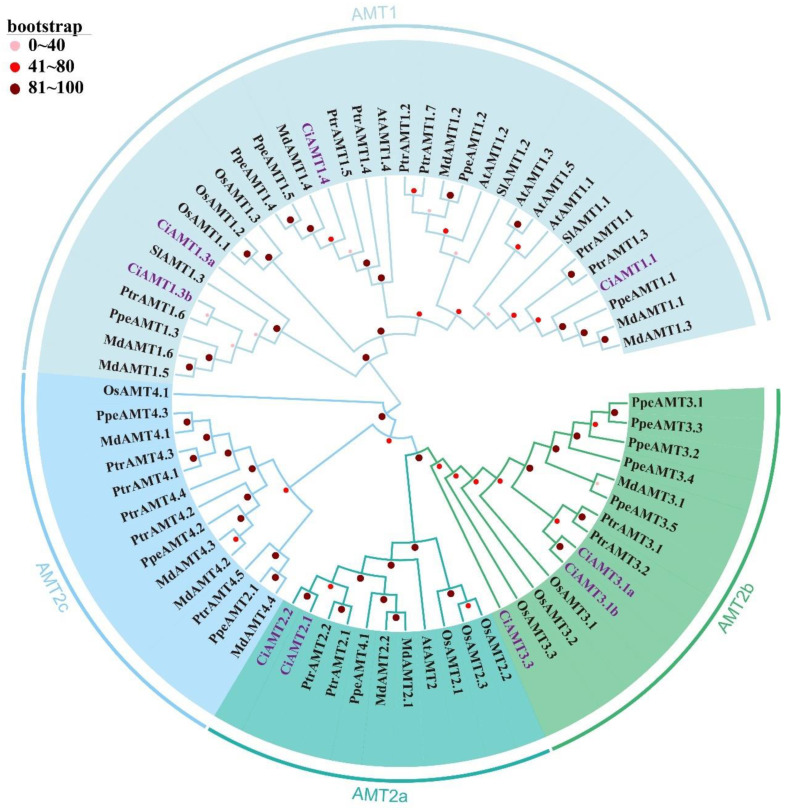
Phylogenetic analysis of *AMT* gene family in pecan (*Carya illinoinensis*), poplar (*Populus trichocarpa*), apple (*Malus domestica*), peach (*Amygdalus persica* L.), tomato (*Lycopersicon esculentum* Miller), rice (*Oryza sativa* L.) and *Arabidopsis* (*Arabidopsis thaliana*). AMT proteins from seven species were divided into two subfamilies (AMT1 and AMT2). The AMT2 was further divided into three clusters (AMT2a, AMT2b, and AMT2c).

**Figure 2 ijms-23-13314-f002:**
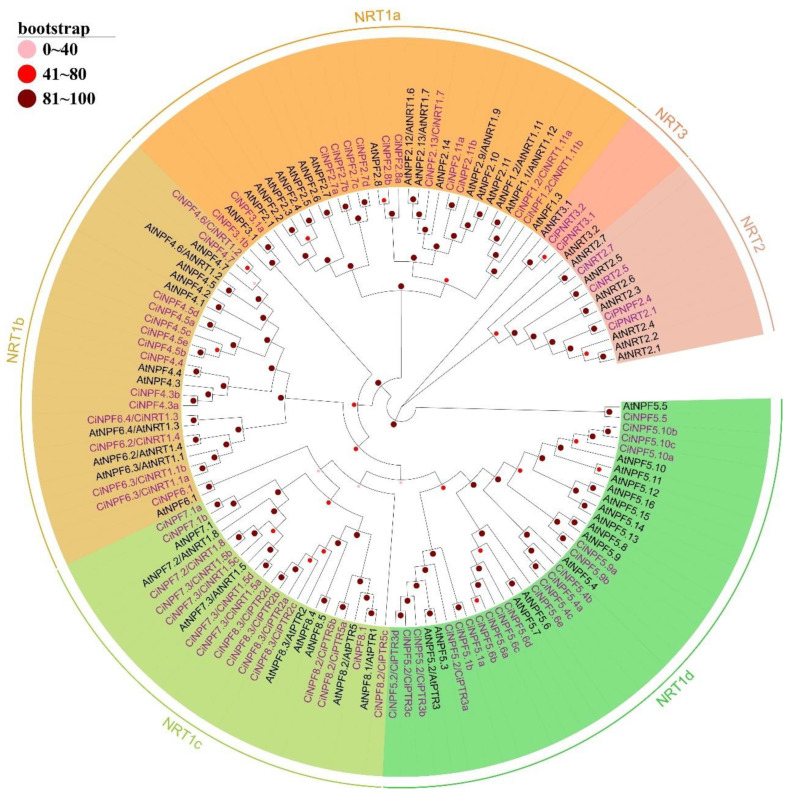
Phylogenetic analysis of *NRT* gene family in pecan (*Carya illinoinensis*) and *Arabidopsis* (*Arabidopsis thaliana*). NRT proteins from two species were divided into three subfamilies (NRT1, NRT2 and NRT3). The NRT1 was further divided into four clusters (NRT1a, NRT1b, NRT1c, and NRT1d).

**Figure 3 ijms-23-13314-f003:**
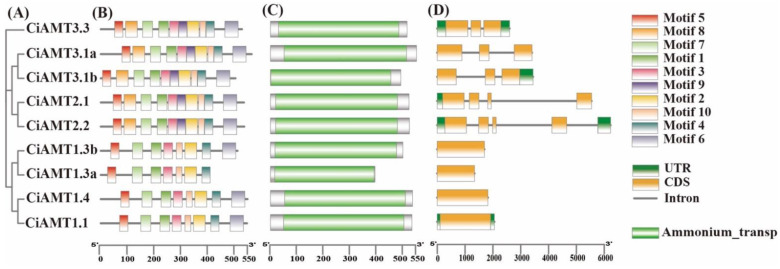
Phylogenetic analysis, conserved motif, conserved domain and gene structural of the *AMT* genes in pecan. Phylogenetic analysis of the *AMT* genes in pecan (**A**). The conserved motifs of the *AMT* genes in pecan (**B**). The conserved domain of the *AMT* genes in pecan (**C**). The gene structure of the *AMT* genes in pecan (**D**).

**Figure 4 ijms-23-13314-f004:**
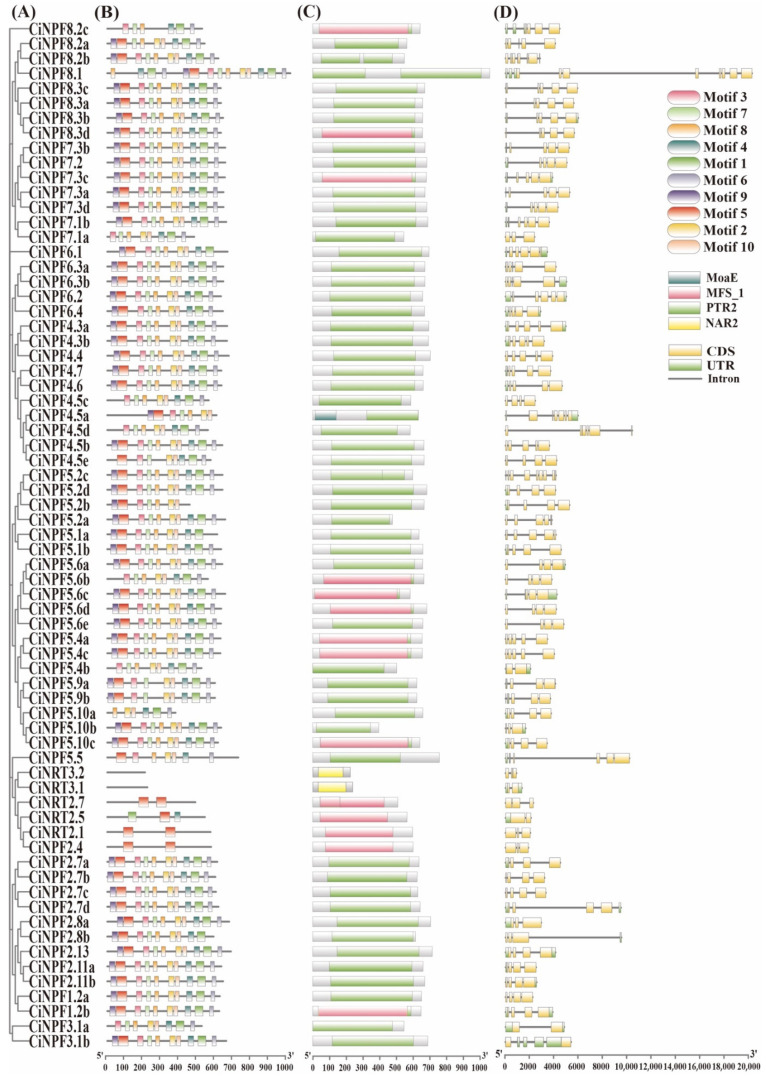
Phylogenetic analysis, conserved motif, conserved domain and gene structural of the *NRT* genes in pecan. Phylogenetic analysis of the *NRT* genes in pecan (**A**). The conserved motifs of the *NRT* genes in pecan (**B**). The conserved domain of the *NRT* genes in pecan (**C**). The gene structure of the *NRT* genes in pecan (**D**).

**Figure 5 ijms-23-13314-f005:**
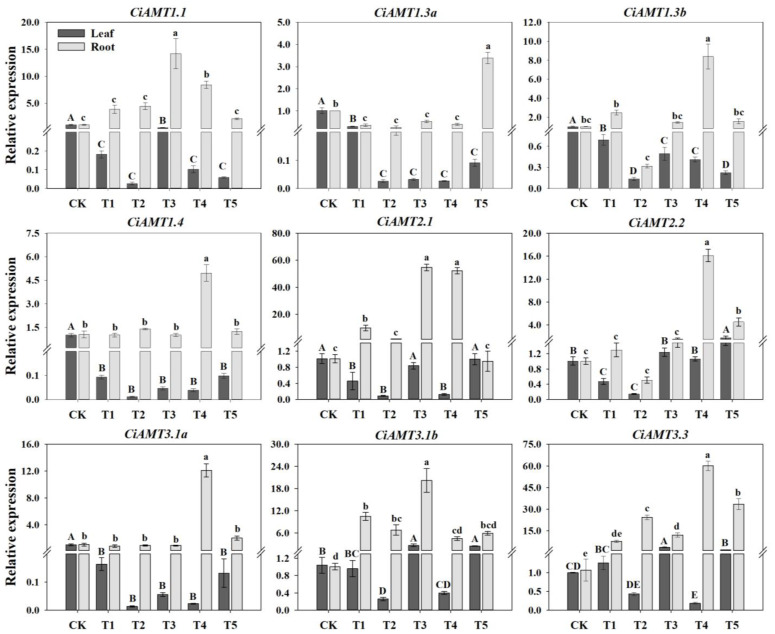
Relative expression of pecan *CiAMT* genes under varying NH_4_^+^:NO_3_^−^ ratios. The expression levels of *CiAMT* in pecan leaves and roots after varying NH_4_^+^:NO_3_^−^ ratio treatments were quantified by qRT-PCR, with *Actin* as the reference gene. Different capital letters indicate significant differences in leaves (*p* < 0.05), and different lowercase letters indicate significant differences in roots (*p* < 0.05).

**Figure 6 ijms-23-13314-f006:**
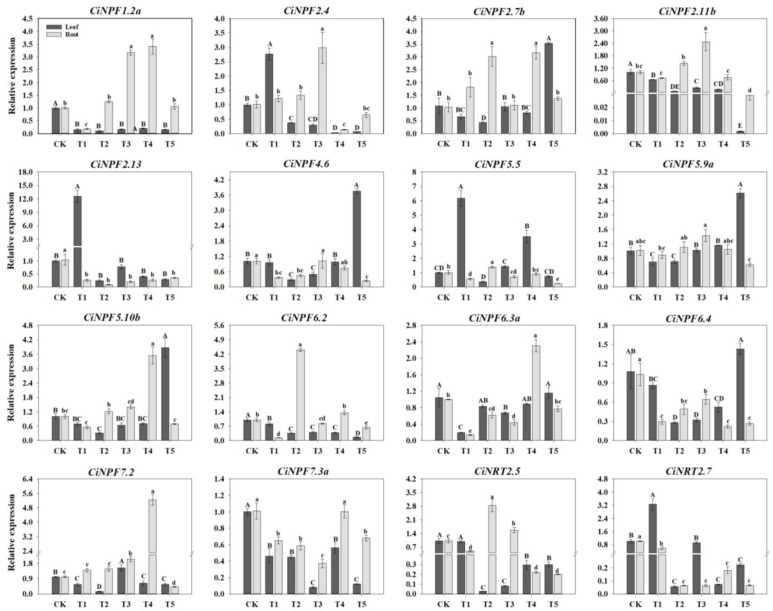
Relative expression of selected pecan *CiNRT* genes under varying NH_4_^+^:NO_3_^−^ ratios. The expression levels of selected *CiNRT* in pecan leaves and roots after varying NH_4_^+^:NO_3_^−^ ratio treatments were quantified by qRT-PCR, with *Actin* as the reference gene. Different capital letters indicate significant differences in leaves (*p* < 0.05), and different lowercase letters indicate significant differences in roots (*p* < 0.05).

**Figure 7 ijms-23-13314-f007:**
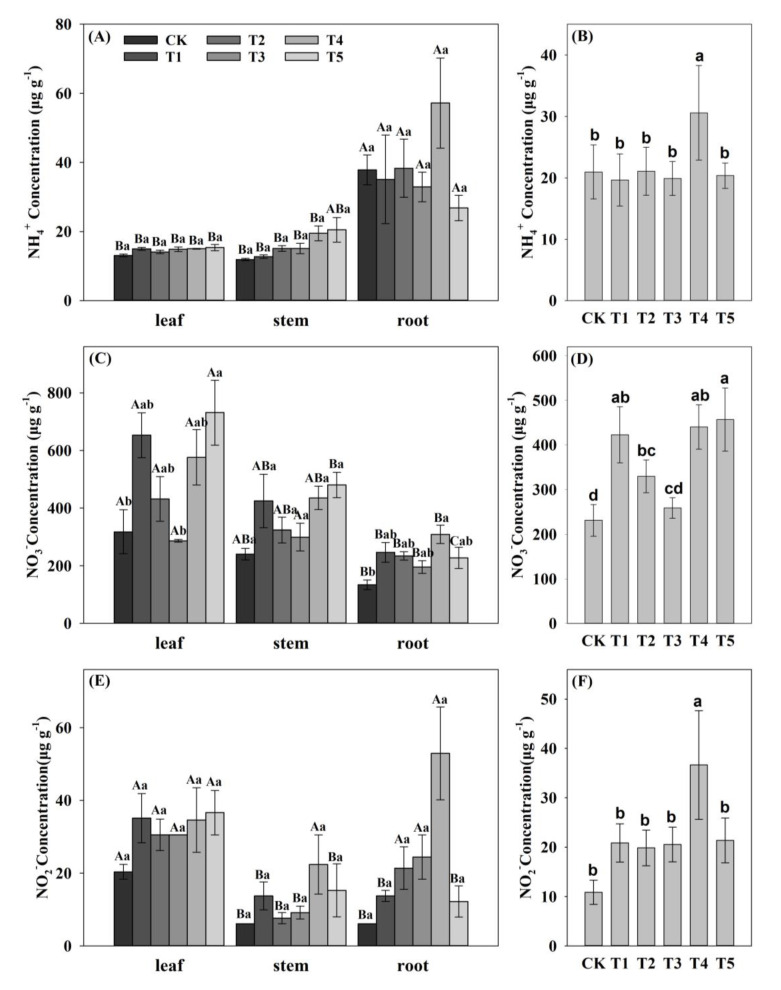
Differences of NH_4_^+^, NO_3_^−^, and NO_2_^−^ concentrations of pecan under varying NH_4_^+^:NO_3_^−^ ratios. Differences of NH_4_^+^ concentrations in pecan leaves, stems, and roots under varying NH_4_^+^:NO_3_^−^ ratios (**A**). The mean values of NH_4_^+^ concentrations in pecan leaves, stems, and roots under varying NH_4_^+^:NO_3_^−^ ratios (**B**). Differences of NO_3_^−^ concentrations in pecan leaves, stems, and roots under varying NH_4_^+^:NO_3_^−^ ratios (**C**). The mean values of NO_3_^−^ concentrations in pecan leaves, stems, and roots under varying NH_4_^+^:NO_3_^−^ ratios (**D**). Differences of NO_2_^−^ concentrations in pecan leaves, stems, and roots under varying NH_4_^+^:NO_3_^−^ ratios (**E**). The mean values of NO_2_^−^ concentrations in pecan leaves, stems, and roots under varying NH_4_^+^:NO_3_^−^ ratios (**F**). Upper capital letters indicate significant differences between organs (*p* < 0.05), and lowercase letters indicate significant differences between varying NH_4_^+^:NO_3_^−^ ratios (*p* < 0.05).

**Figure 8 ijms-23-13314-f008:**
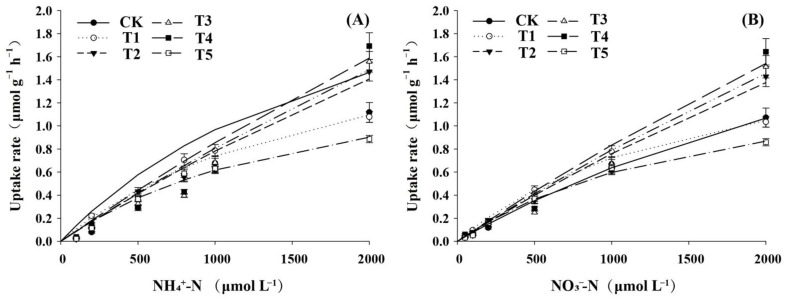
NH_4_^+^ and NO_3_^−^ uptake rates of pecan under varying NH_4_^+^:NO_3_^−^ ratios. (**A**). NH_4_^+^ uptake rates of pecan under varying NH_4_^+^:NO_3_^−^ ratios (**B**). NO_3_^−^ uptake rates of pecan under varying NH_4_^+^:NO_3_^−^ ratios.

**Table 1 ijms-23-13314-t001:** N uptake kinetics of pecan roots under varying NH_4_^+^:NO_3_^−^ ratios. Lowercase letters indicate significant differences between varying NH_4_^+^:NO_3_^−^ ratios (*p* < 0.05).

Treatment	NH_4_^+^-N				NO_3_^−^-N			
	V_max_/ (μmol·g^−1^·h^−1^)	K_m_/ (mmol·L^−1^)	V_max_/K_m_	Goodness of Fit (R^2^)	V_max_/ (μmol·g^−1^·h^−1^)	K_m_/ (mmol·L^−1^)	V_max_/K_m_	Goodness of Fit (R^2^)
CK	3.00 ± 0.21 d	3.40 ± 0.07 d	0.89	0.975	3.30 ± 0.01 c	4.18 ± 0.47 c	0.81	0.996
T1	2.11 ± 0.00 d	1.85 ± 0.18 e	1.17	0.977	1.90 ± 0.02 d	1.62 ± 0.14 d	1.19	0.994
T2	6.96 ± 0.42 c	7.89 ± 0.21 c	0.88	0.966	7.33 ± 0.52 b	8.65 ± 0.18 b	0.85	0.979
T3	9.52 ± 0.93 b	10.80 ± 0.43 b	0.88	0.911	9.05 ± 0.78 a	10.43 ± 0.36 a	0.86	0.954
T4	11.13 ± 0.93 a	12.01 ± 0.39 a	0.92	0.942	10.10 ± 0.55 a	11.10 ± 0.35 a	0.91	0.976
T5	1.68 ± 0.02 d	1.72 ± 0.10 e	0.98	0.978	1.60 ± 0.01 d	1.68 ± 0.12 d	0.97	0.995

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
