# Peer review of "Genome-Wide Identification and Expression Analysis of AMT and NRT Gene Family in Pecan (Carya illinoinensis) Seedlings Revealed a Preference for NH4+-N"

_ijms, 2022, doi:10.3390/ijms232113314_

Round 1

Reviewer 1 Report

In this manuscript, Chen et al. investigate the breadth of two gene families (AMT and NRT) in pecan (Carya illinoinensis), the products of which are involved in the NH4+ and NO3- intake. They also test the effect of the ratio of those ions relative to the intake efficiency and they assay the expression of the AMT and NRT variants in several plant tissues.

This is a well written manuscript that provides valuable insight into an economically important plant species. Please find my comments below:

L. 12-14: Please rephrase. It is difficult to make sense.

L. 50: Please replace "obtain" with a more appropriate word (i.e. uptake/absorption)

L. 50: Pls, replace "was" with "is".

L. 55: Pls, replace "showed" with "show".

L. 56, 90: Pls, replace "played" with "play".

L. 60, 61, 63, 71: Pls, replace "was" with "is".

L. 62: Pls, replace "mediated" with "mediate".

L. 64, 71, 77, 78, 84, 89: Pls, replace "were" with "are".

L. 74: Pls, rephrase ["The transport of NO3- is involved by four protein families"] for better English.

L. 102: Pls, capitalize "w" in "we".

L. 139. Pls, describe the method(s) used for tree reconstructions (i.e. NJ, ML, MP).

L. 170: Pls, state the replication numbers of qPCR samples.

Reviewer 2 Report

I have the following small questions:

1.What is the indoor aerosol incubation?why is the expression patterns of AMT and NRT genes and the uptake characteristics of NH4+ and NO3- in pecan analyzed by indoor aerosol incubation?

2.line 54:The AMT family can usually be divided into two subfamilies, namely AMT1 and AMT2..Is this true for all species? The references do not explain the problem.

3.line 74:"The transport of NO3- is involved by four protein families including ...",NRT3 is not mentioned here.

4.line 223:"Most AMTs (10/11) and NRTs (56/69) were stable."What do you mean, stable? And line 216, 10 were identified as AMT gene family members.(10/11)?

5.You can mark which family the gene belongs to in front of the gene in the phylogenetic trees .

6.The synteny analysis results can be represented by a graph.

7.Why are only roots and leaves selected for gene expression pattern analysis?which can not fully explain the tissue specificity of genes.

8.In Fig7, please explain what is the mean level.

Round 2

Reviewer 2 Report

Thank you for your answer to my question. I learned a lot from this article.